# Fractionation and Lability of Phosphorus Species in Cottonseed Meal-Derived Biochars as Influenced by Pyrolysis Temperature

**DOI:** 10.3390/molecules29020303

**Published:** 2024-01-06

**Authors:** Mingxin Guo, Zhongqi He, Jing Tian

**Affiliations:** 1Department of Agriculture and Natural Resources, Delaware State University, Dover, DE 19901, USA; 2United States Department of Agriculture Agricultural Research Service (USDA-ARS), Southern Regional Research Center, 1100 Allen Toussaint Blvd., New Orleans, LA 70124, USA; zhongqi.he@usda.gov; 3College of Chemistry and Materials Science, Sichuan Normal University, Chengdu 610066, China; wltian@sicnu.edu.cn

**Keywords:** orthophosphate, carbonization, transformation, extraction, bioavailability, mobility

## Abstract

Defatted cottonseed meal (CSM), the residue of cottonseeds after oil extraction, is a major byproduct of the cotton industry. Converting CSM to biochar and utilizing the goods in agricultural and environmental applications may be a value-added, sustainable approach to recycling this byproduct. In this study, raw CSM was transformed into biochar via complete batch slow pyrolysis at 300, 350, 400, 450, 500, 550, and 600 °C. Thermochemical transformation of phosphorus (P) in CSM during pyrolysis was explored. Fractionation, lability, and potential bioavailability of total P (TP) in CSM-derived biochars were evaluated using sequential and batch chemical extraction techniques. The recovery of feed P in biochar was nearly 100% at ≤550 °C and was reduced to <88% at 600 °C. During pyrolysis, the organic P (OP) molecules predominant in CSM were transformed into inorganic P (IP) forms, first to polyphosphates and subsequently to orthophosphates as promoted by a higher pyrolysis temperature. Conversion to biochar greatly reduced the mobility, lability, and bioavailability of TP in CSM. The biochar TP consisted of 9.3–17.9% of readily labile (water-extractable) P, 10.3–24.1% of generally labile (sequentially NaHCO_3_-extractable) P, 0.5–2.8% of moderately labile (sequentially NaOH-extractable) P, 17.0–53.8% of low labile (sequentially HCl-extractable) P, and 17.8–47.5% of residual (unextractable) P. Mehlich-3 and 1 M HCl were effective batch extraction reagents for estimating the “readily to mid-term” available and the “overall” available P pools of CSM-derived biochars, respectively. The biochar generated at 450 °C exhibited the lowest proportions of readily labile P and residual P compounds, suggesting 450 °C as the optimal pyrolysis temperature to convert CSM to biochar with maximal P bioavailability and minimal runoff risk.

## 1. Introduction

Worldwide, approximately 27.4 million tons (equivalent to 121 million bales; 1 ton = 1000 kg) of cotton are produced each year, yielding 11.0 million tons of lint and 16.4 million tons of cottonseed [1,2]. On a dry matter basis, cottonseed contains 14–24% (on average, 17.5%) of oil and 17–33% (on average, 23.0%) of crude protein [3]. The material is commonly processed through mechanical pressing and solvent extraction to recover cottonseed oil [4]. The residue after oil extraction (defatting) is named defatted cottonseed meal (CSM)—a brown, granular organic solid rich in protein, dietary fiber, and nitrogen (N), phosphorus (P), and potassium (K) nutrients. Dry CSM contains 30–50% crude protein, 13–18% dietary fiber, 6.0–9.5% N, 1.1–1.8% P, and 1.2–2.1% K [5,6]. This byproduct is usually utilized as an animal feed or a soil amendment.

In the U.S., 80% of the domestic cottonseed produced is crushed for oil extraction, resulting annually in 1.3 million tons of CSM [7,8]. Commonly available as an organic fertilizer with the nutrient value label “6-2-2”, CSM boosts the growth of lawn, turf, and garden plants if appropriately applied. In the field, however, raw CSM is readily mineralized and the soil-amending effect disappears in one to two growing seasons. Cottonseed meal is also a potential feedstock for valuable biofuels and bioproducts [9]. Converting CSM to biochar and bio-oil via the thermochemical technique “pyrolysis” has been explored [10,11,12]. In a previous work, we applied attenuated total reflection Fourier transform infrared (ATR FT-IR) spectroscopy and solid state ^13^C nuclear magnetic resonance (NMR) spectroscopy to characterize raw CSM and its derived biochar products [13]. While these spectroscopic data provided insights on the changes to the carbon functional groups of CSM during pyrolysis, the lability and phytoavailability of nutrients, particularly P species, in CSM-derived biochar products have not been evaluated.

Biochar is a promising soil amendment, owing to its high environmental recalcitrance (stability) and relatively high water- and nutrient-holding capacities [14]. Agricultural application of CSM-derived biochar in place of raw CSM is expected to achieve persistent soil health improvement effects through ameliorating the soil’s physical, chemical, and biological properties [15]. Nevertheless, the inherent NPK nutrients in CSM may undergo drastic chemical transformations during the pyrolytic conversion process and become less phytoavailable in the resulting biochar. Research indicates that the existing forms and bioavailability of nutrients in biochar are greatly influenced by the pyrolysis conditions, especially the pyrolysis temperature. For example, pyrolytic processing of poultry litter (PL) at 300 °C recovered nearly 82% of the feed N in biochar; the N recovery rate decreased to less than 5% when the pyrolysis temperature was elevated to 500 °C [16]. Concurrently in the pyrolytic process, organic P in PL was transformed into inorganic species and water-soluble P transformed into less labile forms, with the transformation becoming more complete as the pyrolysis temperature increased within 300–800 °C [17,18]. In PL-derived biochars, P was dominated by inorganic phases of amorphous, semicrystalline, and crystalline P minerals, with minor presence of organic P in char components and organo–mineral complexes [17]. The extractability of P in PL-derived biochars using different extraction solutions followed the order: deionized water < Mehlich-3 (200 mM CH_3_COOH + 15 mM NH_4_F + 13 mM HNO_3_ + 1 mM ethylenediaminetetraacetic acid (EDTA) + 250 mM NH_4_NO_3_, pH 2.5) < NaOH–EDTA (250 mM NaOH + 5 mM EDTA) < oxalate (200 mM oxalate, pH 3.5); while in cottonseed hull-derived biochars the order changed to Mehlich-3 < oxalate ≈ deionized water < NaOH–EDTA [17]. Sun et al. [19] also observed the predominance of inorganic P (e.g., Ca-bound phosphates) relative to other P forms in corn stover- and swine manure-derived biochars. Compared with corn stover-derived biochars, swine manure-derived biochars had higher content yet less labile P; and the extractability of P in the biochars was significantly influenced by the acidity of the extraction solution [19]. In soils, the release of P from PL-derived biochar was greatly impacted by the soil pH [20]. As manures generally demonstrate a much higher mineral ash content relative to plant residues [9], the inherent mineral components—especially those multivalent cations such as Ca^2+^, Mg^2+^, Fe^3+^, and Al^3+^—may react with the feed P to form low-solubility phosphate minerals during the pyrolytic process and, subsequently, alter the extractability and phytoavailability of P in biochar products. Considering that CSM has a unique mineral ash composition profile in addition to the relatively high P content, the P speciation and lability (in terms of extractability and phytoavailability) of CSM-derived biochar may be distinctive and respond differently to the effects of the pyrolysis temperature. The main objectives of the present study were to investigate the chemical transformation of P in CSM during pyrolysis; to evaluate the lability of P in CSM-derived biochars generated at varied pyrolysis temperatures; and to gain information regarding optimal pyrolysis conditions for converting CSM into biochar products with desirable P supply dynamics in agricultural and environmental applications. 

## 2. Results and Discussion

### 2.1. General Characteristics of CSM-Derived Biochars

The CSM-derived biochars generated at 300 °C, 350 °C, 400 °C, 450 °C, 500 °C, 550 °C, and 600 °C are referred to as C300, C350, C400, C450, C500, C550, and C600, respectively. Slow pyrolysis of CSM at 300–600 °C resulted in 32.2–53.3% of the dry feed mass remaining as biochar, with the biochar yield decreasing gradually as the pyrolysis temperature increased (Table 1). Meanwhile, the volatile matter content in the biochar products decreased from 43.9% to 9.3%, yet the fixed carbon (C) content increased from 43.7% to >70.0% (Table 1). A lower volatile matter content indicates greater thermal stability in charcoal [21], suggesting that CSM-based biochars generated at higher pyrolysis temperatures are more environmentally recalcitrant, which is consistent with existing observations of PL-derived biochars [16]. It is noteworthy that C550 possessed the greatest fixed C content (70.8%), slightly greater than that of C600 (70.1%) (Table 1). The fixed C yield (fixed C × biochar yield), however, peaked at 25.2% in C400, suggesting 400 °C as the optimal pyrolysis temperature for converting CSM to biochar with maximal stable C storage. Song and Guo [16] also recommended 400 °C pyrolysis to produce biochar from PL for desirable N and stable C recoveries. On a dry mass basis, the raw CSM contained 46.5% organic carbon (OC) and 6.57% mineral ash. The derived biochars showed an OC content of 25.1–55.6% and a mineral ash content of 12.4–20.6%, with the former decreasing while the latter increased as the pyrolysis temperature was elevated in the range of 300–600 °C (Table 1). Considering the biochar yield, the feed OC recovery was estimated at 17.9–65.6%, decreasing as the pyrolysis temperature increased. In comparison, nearly all the ash minerals were preserved in biochar. Owing to their relatively high ash contents, CSM-derived biochars exhibited a high pH value (9.1–10.4) and a high electrical conductivity (EC) level (2.20–3.53 dS m^−1^) (Table 1). Both the pH and the EC of the CSM-derived biochars were elevated as the pyrolysis temperature increased from 300 °C to 450 °C; further increasing the temperature had little influence on the pH and even decreased the EC of the biochar products (Table 1), possibly due to reduced water solubility of the mineral ash components. In comparison, PL-derived biochars subjected to 300–600 °C slow pyrolysis showed pH ranging from 9.5 to 11.5 and EC from 22.8 to 31.0 dS m^−1^ [16]. The reported pH values for biochar products generated from various feedstocks and carbonization conditions ranged from 5.4 to 12.4, and mineral ash content ranged from 1.1% to 82.0% [22].

The raw CSM contained total nitrogen (TN) at 72.3 g kg^−1^. Slow pyrolysis at relatively low temperatures (i.e., ≤350 °C) resulted in slight enrichment of TN in biochar. The TN content of the biochar products decreased steadily as the pyrolysis temperature increased, and was measured at 89.8 g kg^−1^ for C300, 53.3 g kg^−1^ for C450, and 42.1 g kg^−1^ for C600 (Table 1). Significant N losses from thermochemical conversion of N-rich feedstock to biochar may be a concern. Song and Guo [16] noticed that slow pyrolysis of PL at higher temperature (i.e., ≥500 °C) recovered merely <5% of the feed N in biochar. The total P (TP) content of the CSM-derived biochars increased from 23.7 g kg^−1^ for C300 to 37.3 g kg^−1^ for C550 when elevating the pyrolysis temperature from 300 °C to 550 °C, demonstrating ~100% P recovery; further elevating the temperature to 600 °C, however, caused an evident decrease in TP in the biochar product (34.6 g kg^−1^ for C600; TP recovery 87.7%; Table 1). Likely, a portion of the P in CSM was lost via volatilization at temperatures greater than 550 °C. Remarkable portions of the S in CSM were also lost during pyrolysis, whereas nearly all the metal elements were preserved. Pyrolytic treatment of PL at 300–600 °C also showed ~100% recovery of the feed ash minerals and metal elements in biochar [16,20].

The scanning electron microscope (SEM) images of the CSM-derived biochars are shown in Figure 1, illustrating the irregular, porous surfaces of the biochar products. The pores consisted of micropores (5–30 µm) and ultramicropores (0.1–5 µm), with an increased presence of larger, deeper pores in products from higher-temperature pyrolysis. More interestingly, C500 possessed “large” micropores more abundantly than the other biochars (Figure 1). The porosity and the pore size distribution are vital factors determining the capabilities of biochar for adsorbing and retaining water, nutrients, and contaminants [22,23,24]. The adsorption capacity and kinetics of CSM-derived biochars for various environmental media deserve comprehensive investigations in the future. 

### 2.2. Thermochemical Transformation of P in CSM during Pyrolysis

Generally, P compounds exist in biomass materials as both organic phosphate esters (e.g., inositol phosphates, adenosine phosphates, nucleotides, sugar phosphates, phospholipids, and phosphoproteins) and inorganic phosphate salts (e.g., dicalcium phosphate (CaHPO_4_), amorphous tricalcium phosphate (Ca_3_(PO_4_)_2_), octacalcium phosphate (Ca_8_H_2_(PO_4_)_6_∙5H_2_O), pyrophosphates ((P_2_O_7_)^4−^), and polyphosphates ((P_n_O_3n+1_)^(n+2)−^, n > 2)) [25]. In cottonseed and CSM, more than 2/3 of the TP is composed of organic species, with the majority being K/Mg phytates [26,27]. The raw CSM contained 12.7 g kg^−1^ TP (Table 1), of which 7.6% belonged to the inorganic P (IP) category, and 92.4% to organic P (OP) (Table 2). During pyrolysis, nearly all the feed P was preserved (except for at 600 °C, Table 1), yet the IP proportion increased to 56.1–89.2% while the OP proportion decreased to 10.8–43.9% in the derived biochars (Table 2), suggesting the thermochemical transformation of OP into IP. The proportion of IP (operationally defined as 1 M HCl-extractable) in biochar increased initially as the pyrolysis temperature was elevated from 300 °C, peaked at 450 °C (89.2% of TP in C450), reduced gradually as the pyrolysis temperature further increased, and then remained at a relatively stable level (~56%) at ≥550 °C (Table 2). A similar trend of P transformation was observed when PL (TP 13.7 g kg^−1^ consisting of 67.6% of IP and 32.4% of OP) was converted to biochars via slow pyrolysis, with the highest proportion (99.7%) of IP detected in the 450 °C product [18]. At a pyrolysis temperature of 500 °C and above, recalcitrant, less extractable phosphate minerals (e.g., hydroxyapatite and oxyapatite) formed with evident detection in PL-derived biochars, leading to the apparently “decreasing” proportion of IP estimated using 1 M HCl extraction and the “increasing” proportion of OP calculated from the differences between TP and IP in the biochar products [18]. In the present study, the decreased extractability of ash minerals in CSM-derived biochars at higher pyrolysis temperatures (i.e., C500, C500, and C600) is evidenced by the increasing mineral ash contents yet decreasing EC values (Table 1). It is postulated that most (e.g., >90%) of the OP in CSM was transformed into IP during pyrolysis at ≥500 °C. Transformation of OP into IP was also observed during pyrolytic processing of manures, sewage sludge, and other solid biomass residues into biochar [28]. 

### 2.3. Lability-Based Chemical Fractionation of P in CSM-Derived Biochars 

The sequential extraction technique assumes that P exists in soil and other solid media in distinct moieties and associations that demonstrate varying levels of lability, which can be assessed using different extraction reagents in a sequential order [29]. The extractants water, NaHCO_3_, NaOH, and HCl—all used in the present study—were determined to recover readily labile P, generally labile P (adsorbed on crystalline surfaces), moderately labile P (associated with carbonates and Fe/Al oxides or in organic particulates), and low labile P (bound in Ca-minerals), respectively, in soils and manures through sequential extraction [30,31,32]. The readily to moderately labile portions of P that are sequentially extractable by water, NaHCO_3_, and NaOH are broadly categorized as “labile,” readily to mid-term plant-available, and subject to runoff, leaching, and other environmental processes. The sequentially HCl-extractable P is generally viewed as “non-labile” but may be plant-available following long-term (e.g., >5 years) field weathering [33]. The unextractable, residual fractions are largely inert and plant-unavailable. As shown in Figure 2, nearly all the TP in raw CSM was labile, of which 87.9% was water-extractable, 7.5% NaHCO_3_-extractable, 4.6% NaOH-extractable, and <0.05% HCl-extractable. In CSM-derived biochars, residual P became significant, accounting for 17.8–47.5% of the TP. Meanwhile, water-extractable P decreased to 17.9% in C300, further decreased with an elevated pyrolysis temperature to 9.3% in C450, and then remained relatively stable at around 10% in C500, C550, and C600 (Figure 2). The proportion of NaHCO_3_-extractable P increased slightly to 10.5% in C300, 18.6% in C450, and 24.1% in C600. The proportion of NaOH-extractable P was constantly low, ranging between 0.4–2.8% in all the biochar products generated at different pyrolysis temperatures (Figure 2). On the contrary, the sequentially HCl-extractable P fractions became significant, comprising 45.1% of the TP in C300, increasing with the rising pyrolysis temperature to 53.8% in C450, and then decreasing gradually to 17.0% in C600 (Figure 2). Overall, the conversion to biochar remarkably reduced the lability of TP in CSM, with the effect more magnificent at higher pyrolysis temperatures. The conversion of PL to biochar achieved similar results, with water-soluble P decreasing its proportion from 49.5% in raw PL to 11.7% in 300 °C biochar and 2.4% in 600 °C biochar [18].

All three species of P—dissolved inorganic orthophosphate P (P_ir_), dissolved inorganic polyphosphate P (P_ix_), and dissolved organic phosphorus (P_o_)—were detected in the labile (sequentially extractable by water, NaHCO_3_, and NaOH) P in raw CSM and the derived biochars. The most labile, water-soluble P in raw CSM, for example, was composed of 10.9% P_ir_, 20.3% P_ix_, and 68.8% P_o_. The proportions for the biochars shifted to 28.9–36.3%, 57.7–67.9%, and 2.3–6.0%, respectively, without clear trends against the pyrolysis temperature (Table 3). The predominance of P_ix_ in biochar’s water-soluble P suggests the inadequacy of metal elements (primarily K, Na, Ca, and Mg) in CSM to form orthophosphate salts with additional inorganic P induced from the thermochemical transformation of OP (∑(Na + K + Ca + Mg):OP equivalent ratio = 0.97, Table 1). Other anionic components (e.g., Cl^−^, SO_4_^2−^, CO_3_^2−^, and HCO_3_^−^) co-existed in biochars and might compete with phosphates for salt-forming metal ions. Previous research revealed the predominance of P_ir_ instead of P_ix_ in water-extractable P from PL-derived biochars, presumably owing to the high ash content (28.5 wt%) and ∑(Na + K + Ca + Mg):P equivalent ratio (7.45) of PL [18,20]. In soil, P_ix_ is readily hydrolyzable to become the plant-available P_ir_ form [34]. The sequentially NaHCO_3_-extractable P from raw CSM consisted of 37.4% P_o_, 49.8% P_ix_, and 12.8% P_ir_. The P_o_ proportion decreased to 13.9% in C300 and remained at 2.5–5.3% in the products of higher pyrolysis temperatures; the P_ix_ proportion increased slightly to 53.2% in C300 and 55.8% in C350, and then decreased gradually in the higher pyrolysis temperature biochars to ~24.5% in C550 and C600; whereas the P_ir_ proportion increased steadily in higher pyrolysis temperature biochars, from 32.9% in C300 to >72.0% in C550 and C600 (Table 3). The sequentially NaOH-extractable P in raw CSM still demonstrated a mixed composition of P_o_ (54.4%), P_ix_ (37.3%), and P_ir_ (8.4%). Nevertheless, P_o_ disappeared from this particular P fraction in the derived biochars, with dissolved inorganic P species (consisting of ~40% P_ir_ and ~60% P_ix_) as the sole components (Table 3). Given the great potency of the strongly alkaline NaOH solution for extracting constituents from solid organic matrices, the null presence of P_o_ indicated that roughly all biochar OP was recovered through the prior extractions with water and NaHCO_3_.

Cumulatively, P_ir_, P_ix_, and P_o_ constituted 10.9%, 23.3%, and 65.8% of the total extractable (including sequentially HCl-extractable) P, respectively, in raw CSM. For the derived biochars, the proportions changed to 73.3–82.8%, 15.7–24.7%, and 0.32–2.93%, respectively (Figure 3). The P_ir_ proportion increased while the P_ix_ proportion decreased for biochars when elevating the pyrolysis temperature in the range of 300–450 °C; the trends were reversed, however, with the further increase of the temperature in the range of 450–600 °C (Figure 3). Likely, OP was first transformed into polyphosphates and then further into orthophosphates during pyrolysis, with both steps promoted by increasing the pyrolysis temperature [18]. At temperatures >450 °C, additional polyphosphates from OP transformation could not form orthophosphates due to the limited availability of metal elements in CSM, and they therefore accumulated in biochar products. Overall, the total extractable P in CSM-derived biochars was dominated by inorganic species (i.e., >97%), with organic forms as a minor component (i.e., <3%) (Figure 3). 

### 2.4. Batch Extractability-Based Lability and Bioavailability of P in CSM-Derived Biochars

The mobility, lability, and bioavailability of P in soil and amendments have commonly been assessed through batch extraction with water and formulated solutions [35]. The Olsen solution (0.5 M NaHCO_3_, pH 8.5), for instance, is commonly used to estimate the phytoavailable P pool in neutral, alkaline, and calcareous soils. The Bray-1 extractant (0.025 M HCl in 0.03 M NH_4_F) is expected to recover seasonally available P in water-soluble and adsorbed forms in pH <7.5 soils. The Mehlich-3 solution was designed to remove adsorbed and Fe/Al oxides-complexed P from acidic and neutral soils [36]. In the present study, water was able to extract 87.9% of TP in raw CSM, but only 9.3–17.9% in the derived biochars (Figure 4). Water-soluble nutrients are readily available to plants [37]. Clearly, conversion to biochar dramatically reduced the lability and immediate bioavailability of P in CSM, transforming the byproduct into a “slow release fertilizer.” The water extractability of P in CSM-derived biochars decreased gradually from 17.9% to 9.3% of TP as the pyrolysis temperature increased in the range of 300–450 °C; further elevating the pyrolysis temperature reversed the trend and resulted in 10.6% of TP as water-extractable P in C600 (Figure 4), matching the fluctuation patterns of P_ir_ in the extracts (Table 3). Li et al. [18] reported that 2.4–11.7% of the TP in PL-derived biochars was water-extractable, with the proportion decreasing as the pyrolysis temperature increased from 300 °C to 450 °C and above. The Olsen solution extracted 24.0–29.8% of TP from biochars, yet merely 16.0% from raw CSM. The Olsen solution’s P extractability was the lowest for C450, similar to the observations made during water extraction (Figure 4). In comparison, the Bray-1 solution was able to recover 33.9–38.5% of TP from the biochars. The effects of the pyrolysis temperature on the Bry-1 P extractability of CSM-derived biochars, however, were vaguely reflected in random patterns. The more corrosive Mehlich-3 solution extracted 35.5–43.7% of TP from the biochars. Once again, the effects of the pyrolysis temperature displayed no clear trends. The strongly acidic extractant 1 M HCl removed 56.1–89.2% of TP from the biochars, but less than 8.0% from raw CSM. Natural dissolved organic matter condenses and even precipitates at pH <4.0 [38]. The tremendously lower HCl extractability of P in raw CSM compared with that of the derived biochars corroborated the predominance of OP in CSM (Table 2) and the extensive transformation of OP into IP during pyrolysis. 

The outstandingly higher extraction effectiveness of HCl relative to other extractants (Figure 4) suggests the major presence of Ca/Mg-bound P in CSM-derived biochars, due to enriched Ca and Mg contents through the pyrolytic decomposition of organic components (Table 1). When elevating the pyrolysis temperature, the HCl-extractability of P in the biochar products was initially promoted to a maximal level (89.2% of TP) at 450 °C and afterwards gradually decreased to 56.1% (of TP) at 600 °C (Figure 4), possibly due to the formation of recalcitrant P species (e.g., oxyapatite and hydroxyapatite) at higher pyrolysis temperatures [18]. Most of the “nominal” OP in C500, C550, and C600 (Table 2) might belong to HCl-unextractable residual P. Overall, the extraction efficiency of the test reagents for recovering P in CSM-derived biochars followed the order: water < Olsen < Bray-1 ≤ Mehlich-3 < HCl. Mehlich-3 is able to recover readily labile (water-soluble), generally labile (adsorbed), and moderately labile (primarily Fe/Al-bound) P in soil and biochars [18,36], and therefore may serve as an effective reagent for estimating the short- to mid-term plant availability of P in CSM-derived biochars. The HCl extractability tracked the changes in the non-residual P fraction of biochars generated at different pyrolysis temperatures (Figure 2 and Figure 4). Moreover, the low labile, HCl-extractable P (bound to Ca/Mg minerals) is plant-available in the long term [33]. As such, 1 M HCl may be an optimal extractant for evaluating the general bioavailability of P in CSM-derived biochars.

## 3. Materials and Methods

### 3.1. Defatted Cottonseed Meal

Cottonseed meal was obtained from an agricultural facility in Kentwood, LA, USA. The material was received in a commercial 11.4 kg (25 lbs) bag with a label showing minimum protein content 39.0%, minimum fat content 1.0%, and maximum fiber content 14.0%. Laboratory measurements revealed that 99.4% of the CSM mass was in 0.05–4 mm particles. The CSM contained 91.6 wt% dry matter and 8.4 wt% moisture. The dry matter was composed of 6.6 wt% mineral ash and 93.4 wt% organic constituents. On a dry matter basis, the CSM showed an OC content of 451.8 g kg^−1^, TN content of 70.2 g kg^−1^, TP content of 12.7 g kg^−1^, and total K content of 15.5 g kg^−1^. The CSM was used as received. 

### 3.2. Converting CSM to Biochar

Converting CSM to biochar was achieved following the complete batch slow pyrolysis method as described by Song and Guo [16]. Approximately 600 g of CSM was loaded into a 1000-mL metal container (10.5 cm i.d. × 12.0 cm height) and heated in an electricity-powered muffle furnace (Thermo Fisher Scientific, Inc., Suwanee, GA, USA) at a selected peak temperature. A small 5-mm hole was pre-installed in the container lid, allowing vapors (smoke) to escape from the container during pyrolysis. The furnace temperature was increased from the ambient temperature to the selected peak temperature at 20 °C min^−1^ and maintained at the peak value until the pyrolysis reaction (pyrolytic decomposition) in the container was complete, as indicated by no further visible smoke being emitted by the furnace [16]. It took 141–445 min to reach complete pyrolysis, depending on the peak temperature in the range of 300–600 °C. Once complete pyrolysis was achieved, the container was withdrawn from the furnace, immediately sealed in the lid hole with a piece of metal tape, and air-cooled to room temperature. Afterwards, the biochar inside the container was transferred into a Ziploc plastic bag and stored at room temperature in a dark cabinet in the laboratory. The yield of the biochar at each peak pyrolysis temperature was computed accordingly using the mass percentage of the dry feed CSM. Duplicate biochar samples were prepared at each of the selected peak pyrolysis temperatures. 

### 3.3. General Characterization of CSM and the Derived Biochars

Aliquots of the raw CSM and its derived biochars were ground to <0.85 mm and analyzed for general properties as a soil amendment. The moisture content and organic matter content were measured through the open heating of 5.0 g of samples at 105 °C and 500 °C, respectively, for 8 h [16]. The volatile matter content and the mineral ash content were determined following the standard methods for the chemical analysis of wood charcoal D1762-84 [39]. The OC content and the TN content were analyzed using a Shimadzu TC/TN analyzer with a solid sampling module (Shimadzu, Kyoto, Japan). The pH and the EC of the samples were measured in 1:5 solid/water (*w*/*w*) 24-h room temperature extracts using an Accumet AB15 pH meter (Fisher Scientific, Suwanee, GA, USA) and an Orion Star A212 conductivity meter (Thermo Scientific, Beverly, MA, USA), respectively. The contents of the elements S, Ca, Mg, K, Na, Fe, and Zn were quantified through the inductively coupled plasma (ICP) measurement (Spectro CirOs, Mahwah, NJ, USA) of acid-digested samples [16]. 

To examine the surface features, in particular, the micropore distribution of CSM-derived biochars generated at different pyrolysis temperatures, biochar samples were coated with a thin film of gold and scanned using a Hitachi S-4700 cold field emission scanning electron microscope (SEM) (Hitachi Inc., Tokyo, Japan). 

### 3.4. Chemical Characterization of P in CSM and the Derived Biochars

Aliquots of the raw CSM and the biochar samples were further ground to <0.15 mm prior to P characterization. The TP content was determined following colorimetric P measurement of acid-digested samples [18]. The TP consists of IP and OP in most environmental matrices. The IP content of the samples was quantified using 24-h 1 M HCl extraction followed by colorimetric P measurement of the extracts [18]. The difference between the TP and IP contents of a sample was computed as the OP content.

The TP of raw CSM and the derived biochars was further fractionated through sequential extraction into different operationally-defined pools to evaluate the lability of various P fractions. The samples were sequentially extracted at room temperature with Milli-Q water, 0.5 M NaHCO_3_, 0.1 M NaOH, and 1 M HCl at a 1:50 solid/solution ratio [18]. The extracts obtained at each sequential extraction step following 24-h rotary shaking, centrifugation, and 0.2-µm syringe filtration of supernatant and MilliQ water washing of pellet were analyzed for concentrations of total dissolved P (P_t_), dissolved organic P (P_o_), and dissolved inorganic P (P_i_) that was further fractionated into dissolved inorganic orthophosphate-P (P_ir_) and dissolved inorganic polyphosphate-P (P_ix_). The concentrations of P_t_, P_i_, and P_ir_ were determined using the phosphomolybdate blue method after acidic K_2_S_2_O_8_ autoclave digestion, H_2_SO_4_ autoclave digestion, and no pretreatment of the sequential extracts, respectively [18]. The P_o_ was calculated as the difference between P_t_ and P_i_ (P_o_ = P_t_ − P_i_), and P_ix_ as the difference between P_i_ and P_ir_ (P_ix_ = P_i_ – P_ir_).

The lability of P in raw CSM and its derived biochars was further assessed using batch extraction techniques. Aliquots of the samples were extracted separately with deionized water, Mehlich-3 solution, Bray-1 solution, Olsen solution, and 1 M HCl at 1:50 solid/solution ratio for 24 h under room temperature rotary agitation [40]. The extracts were passed through 0.2-μm syringe filters and measured for Pt concentrations following the phosphomolybdate blue method after acidic K_2_S_2_O_8_ autoclave digestion [18]. 

### 3.5. Experimental Quality Control and Statistical Data Analysis

All laboratory experiments of sample characterization and analysis were conducted in duplicate (*n* = 2). In the experiments involving P measurements, acid-washed glass or Teflon ware was used. Procedure blanks without sample addition were included. Certified standard P solutions (RICCA Chemical Company, Arlington, TX, USA) were used to establish the calibration curves necessary in P colorimetric measurements and were subject to the same digestion processes as the extract samples. 

Analytical procedure blanks were deducted from sample measurements in data processing. Quantitative results are expressed as the mean ± standard deviation of experimental duplicates (*n* = 2). The relative abundance of different P species in raw CSM and its derived biochars is indicated as a percentage relative to the TP concentration. Variations in the speciation and lability of P in raw CSM and the derived biochars from varying pyrolysis temperatures were statistically evaluated at the level of significance α = 0.05 following the analysis of variance (ANOVA) and the Fisher least significant difference (LSD) methods. 

## 4. Conclusions

Phosphorus existed in raw CSM predominantly (>90%) in organic species and marginally (<10%) in inorganic species. Slow pyrolysis of CSM at ≤550 °C recovered nearly all the feed TP in biochar; the loss of P in pyrolysis vapors occurred at higher temperatures and exceeded 12% of TP at 600 °C. During pyrolysis, OP was transformed into IP, first to polyphosphates and subsequently to orthophosphates as facilitated by higher temperatures and a greater co-presence of metal elements. At temperatures >450 °C, the formation of recalcitrant Ca/Mg phosphate minerals became evident. 

Conversion into biochar significantly reduced the mobility, lability, and bioavailability of P in CSM. Using sequential extraction techniques, the TP in CSM-derived biochars was fractionated into 9.3–17.9% of readily labile (water-extractable) P, 10.3–24.1% of generally labile (NaHCO_3_-extractable) P, 0.5–2.8% of moderately labile (NaOH-extractable) P, 17.0–53.8% of low labile (HCl-extractable) P, and 17.8–47.5% of residual (unextractable) P. Mehlich-3 and 1 M HCl were effective batch extraction reagents that could be used to estimate the “readily to mid-term” available and the “overall” available P pools of CSM-derived biochars, respectively. Among the seven biochars prepared at different pyrolysis temperatures, C450 exhibited the lowest proportions of readily labile P and residual P, implicating 450 °C as the optimal pyrolysis temperature to convert CSM to biochar with maximal P bioavailability and minimal runoff risk.

## Figures and Tables

**Figure 1 molecules-29-00303-f001:**
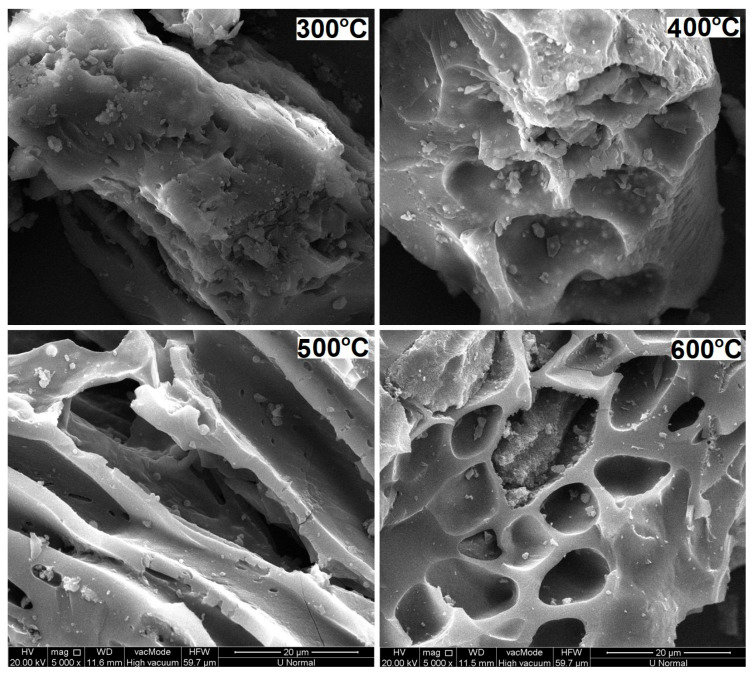
Scanning electron micrographs at a magnification of 5000× of cottonseed meal-derived biochars generated at 300–600 °C slow pyrolysis temperatures.

**Figure 2 molecules-29-00303-f002:**
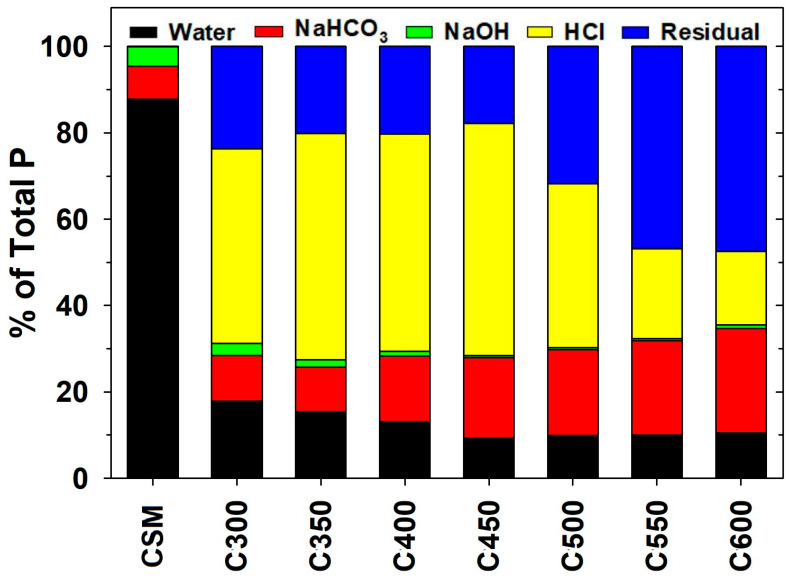
Fractionation of phosphorus (P) in raw cottonseed meal (CSM) and the derived biochars into water-extractable, NaHCO_3_-extractable, NaOH-extractable, HCl-extractable, and residual forms. Data are expressed as the means of experimental duplicates. The coefficients of variation of the experimental duplicates are within 3%.

**Figure 3 molecules-29-00303-f003:**
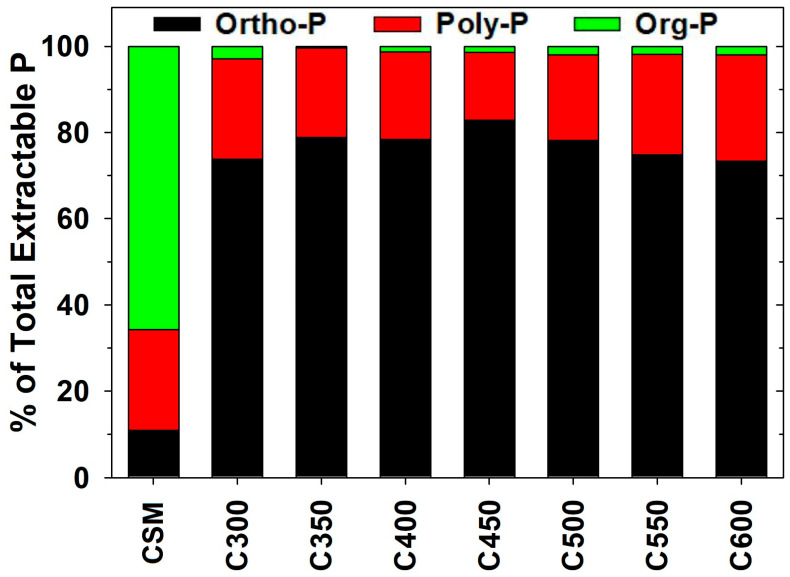
Composition of total extractable phosphorus (P) (sum of sequentially extractable P fractions by water, NaHCO_3_, NaOH, and HCl) in raw cottonseed meal (CSM) and the derived biochars. Data are expressed as the means of experimental duplicates.

**Figure 4 molecules-29-00303-f004:**
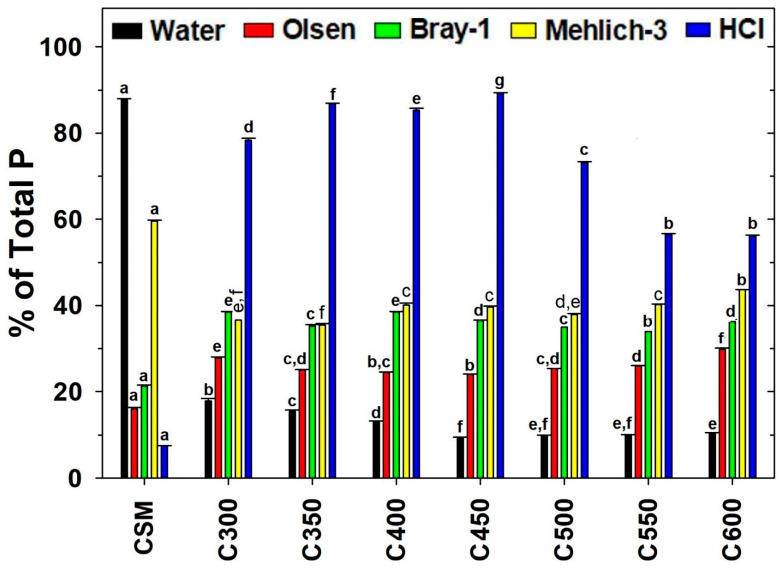
Extractability (indicating lability and potential bioavailability) of phosphorus (P) in raw cottonseed meal (CSM) and the derived biochars by water, Olsen extractant, Bray-1 extractant, Mehlich-3 extractant, and 1 M HCl. Data are expressed as the means of experimental duplicates. Letters above bars denote the significance of differences between treatment levels.

**Table 1 molecules-29-00303-t001:** General characteristics (dry mass-based) of cottonseed meal (CSM)-derived biochars generated through complete slow pyrolysis at different peak temperatures. Data are expressed as means of experimental duplicates with a coefficient of variation <2% if not specified.

	CSM	C300	C350	C400	C450	C500	C550	C600
Pyrolysis T, °C		300	350	400	450	500	550	600
Yield ^‡^, %	100	53.3	46.7	40.8	36.0	35.0	33.4	32.2
Fixed C, %	-	43.7	52.3	61.8	66.2	68.7	70.8	70.1
Volatile matter, %	-	43.9	33.6	22.4	15.6	12.4	9.7	9.3
OC, %	46.5	55.6	36.8	30.2	25.9	25.6	25.3	25.1
Mineral ash, %	6.57	12.4	14.1	15.8	18.2	18.9	19.5	20.6
pH ^†^	-	9.1	9.6	10.1	10.4	10.3	10.3	10.3
EC ^†^, dS m^−1^	-	2.20	2.26	2.99	3.53	3.28	3.39	3.24
TN, g kg^−1^	72.3	89.8	71.7	58.7	53.3	50.1	46.7	42.1
TP, g kg^−1^	12.7	23.7	27.1	31.3	33.9	34.9	37.3	34.6
S, g kg^−1^	4.21	3.04	2.75	2.15	2.05	1.75	1.35	0.98
K, g kg^−1^	15.5	29.6	33.5	37.6	42.4	43.6	46.4	47.7
Na, g kg^−1^	2.27	4.29	4.81	5.57	6.18	6.36	6.62	6.99
Ca, g kg^−1^	2.06	3.85	4.48	5.04	5.70	5.85	6.15	6.37
Mg, g kg^−1^	6.23	12.1	13.4	15.3	17.0	17.6	18.6	19.2
Fe, g kg^−1^	0.087	0.166	0.190	0.215	0.238	0.249	0.257	0.270
Zn, g kg^−1^	0.059	0.109	0.124	0.146	0.163	0.169	0.171	0.188

T: temperature; EC: electrical conductivity; OC: organic carbon. ^†^ in 1:5 solid/water extract; ^‡^ % of dry feed (CSM) mass.

**Table 2 molecules-29-00303-t002:** Contents of inorganic phosphorus (IP, measured by 1 M HCl extraction) and organic phosphorus (OP, calculated from the difference between TP and IP) in cottonseed meal (CSM) and the derived biochars and their fractions relative to total phosphorous (TP). Data are expressed as the means ± standard deviations of experimental duplicates. Superscript letters denote the significance of differences between treatment levels.

	IP		OP	
	g kg^−1^	% of TP	g kg^−1^	% of TP
CSM	0.962 ± 0.003	7.57 ± 0.18 ^a^	11.75 ± 0.023	92.43 ± 0.44 ^a^
C300	18.60 ± 0.12	87.35 ± 0.77 ^d^	5.14 ± 0.18	12.65 ± 1.04 ^d^
C350	23.60 ± 0.048	86.77 ± 0.44 ^d^	3.60 ± 0.12	13.23 ± 0.70 ^d^
C400	26.72 ± 0.14	85.35 ± 1.26 ^d^	4.59 ± 0.39	14.65 ± 1.41 ^d^
C450	30.29 ± 0.048	89.20 ± 0.29 ^d^	3.67 ± 0.098	10.80 ± 0.44 ^d^
C500	25.56 ± 0.071	73.13 ± 0.35 ^c^	9.39 ± 0.12	26.87 ± 0.49 ^c^
C550	21.09 ± 0.024	56.58 ± 0.33 ^b^	16.19 ± 0.12	43.42 ± 0.62 ^b^
C600	19.41 ± 0.071	56.10 ± 1.52 ^b^	15.19 ± 0.53	43.90 ± 1.83 ^b^

**Table 3 molecules-29-00303-t003:** The proportional compositions of sequentially water-extractable, NaHCO_3_-extractable, and NaOH-extractable phosphorus (P) fractions in cottonseed meal (CSM) and the derived biochars. Data are expressed as the means ± standard deviations of experimental duplicates. Superscript letters denote the significance of differences between treatment levels.

	Water-Soluble P			NaHCO_3_-Extractable P			NaOH-Extractable P		
	P_ir_	P_ix_	P_o_	P_ir_	P_ix_	P_o_	P_ir_	P_ix_	P_o_
	% of Water-Soluble P_t_			% of NaHCO_3_-Extractable P_t_			% of NaOH-Extractable P_t_		
CSM	10.9 ± 0.18 ^a^	20.3 ± 0.26 ^a^	68.8 ± 0.22 ^a^	12.8 ± 0.98 ^a^	49.8 ± 1.74 ^c,d^	37.4 ± 1.57 ^a^	8.36 ± 0.33 ^a^	37.3 ± 0.50 ^a^	54.4 ± 0.87 ^a^
C300	36.3 ± 0.34 ^d^	57.7 ± 0.93 ^b^	6.02 ± 1.15 ^b^	32.9 ± 0.29 ^b^	53.2 ± 0.77 ^d^	13.9 ± 0.90 ^b^	40.3 ± 0.36 ^b,c,d^	59.7 ± 2.33 ^b^	0.00 ± 0.00 ^b^
C350	35.7 ± 0.34 ^d^	62.0 ± 0.54 ^c,d^	2.35 ± 0.40 ^b^	41.8 ± 0.69 ^c^	55.8 ± 1.71 ^d^	2.47 ± 1.67 ^c^	42.2 ± 1.04 ^c,d^	57.8 ± 2.57 ^b^	0.00 ± 0.00 ^b^
C400	32.5 ± 0.53 ^c^	64.5 ± 1.83 ^d,e^	2.97 ± 2.03 ^b^	49.5 ± 1.17 ^d^	46.3 ± 1.89 ^c^	4.15 ± 2.00 ^c^	39.9 ± 1.89 ^b^	60.1 ± 4.01 ^b^	0.00 ± 0.00 ^b^
C450	28.9 ± 0.21 ^b^	66.2 ± 0.82 ^e^	4.91 ± 2.90 ^b^	61.4 ± 0.90 ^e^	34.7 ± 2.82 ^b^	3.91 ± 2.89 ^c^	40.5 ± 2.13 ^b,c^	59.5 ± 9.17 ^b^	0.00 ± 0.00 ^b^
C500	29.1 ± 0.21 ^b^	67.9 ± 0.55 ^e^	2.94 ± 0.41 ^b^	62.0 ± 0.49 ^e^	32.7 ± 1.86 ^b^	5.29 ± 2.54 ^c^	35.9 ± 0.61 ^b^	64.1 ± 7.61 ^b^	0.00 ± 0.00 ^b^
C550	30.4 ± 0.58 ^b,c^	67.2 ± 0.76 ^e^	2.49 ± 1.00 ^b^	72.5 ± 0.71 ^f^	24.2 ± 1.49 ^a^	3.37 ± 1.52 ^c^	36.4 ± 1.29 ^b,c^	63.6 ± 8.52 ^b^	0.00 ± 0.00 ^b^
C600	35.9 ± 0.79 ^d^	61.7 ± 1.57 ^c^	2.33 ± 0.79 ^b^	72.0 ± 0.64 ^f^	24.7 ± 1.79 ^a^	3.28 ± 1.87 ^c^	38.9 ± 2.31 ^b,c^	61.1 ± 5.66 ^b^	0.00 ± 0.00 ^b^

P_t_: total dissolved phosphorus (P); P_o_: dissolved organic phosphorus; P_ir_: dissolved inorganic orthophosphate-P; P_ix_: dissolved inorganic polyphosphate-P.

## Data Availability

The data presented in this study are available on request from the corresponding author.

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
