# Peer review of "Fractionation and Lability of Phosphorus Species in Cottonseed Meal-Derived Biochars as Influenced by Pyrolysis Temperature"

_molecules, 2024, doi:10.3390/molecules29020303_

Round 1

Reviewer 1 Report

Comments and Suggestions for Authors

Dear authors!

The MS titled “Fractionation and Lability of Phosphorus Species in Cottonseed Meal-derived Biochars as Influenced by Pyrolysis Temperature” was prepared based on original research. The information presented is of scientific interest and practical significance. In my opinion, it corresponds to the theme of the Molecules Journal. The MS title reflects the content of the article. Generally, the work is easy to understand. However, I cannot recommend MS in the form presented for publication in Molecules because it contains some inaccuracies.

First of all, the structure of the article does not meet the requirements of the Molecules journal. According to the “Instructions for authors” in Molecules, the following sequence of sections is recommended: Introduction, Results, Discussion, Materials and Methods.

Abstract. The abstract provides a fairly clear overview of the research. Information is presented in a logical sequence.

Keywords. It is not recommended to include the words used in the title of MS.

Introduction. The relevance of the problem to which the research is aimed is shown. The rationale for scientific and methodological approaches to solving this problem is presented. However, there is no clear statement of the purpose of the study.

P. 1, L. 32, 33, 43: It is recommended to use more commonly used mass units than megagrams (Mg), especially since the MS repeatedly mentions magnesium with the same symbol.

Materials and Methods. In general, the methodological section is written in sufficient detail, but subsection 2.5 needs to be improved:

P. 4, L. 172. It was shown that all laboratory experiments of sample characterization and analysis were conducted in duplicates. P. 4, L. 179: “Quantitative results are expressed as mean ± standard deviation of duplicate measurements”. What the authors mean by “in duplicates measurements” – two independent experiments, data of which were averaged? How many biological and analytical replicates were in each measurement?

Results and Discussion. The authors presented quite a lot of Figures and Tables, which illustrate the work well. Figure 1 is of particular interest. However, there are a number of comments on the design of the illustrations.

First of all, if the number of repetitions of measurements differed, it is advisable to display this information in Tables and Figures (n = ?).

P. 6, Table 1. The heading of the 1st column (“Product”) does not seem to be correct.

P. 10, Figure 3. Сaption: “superscript letters denote significance of difference between treatment levels” is wrong.

In Figures 2–4 captions, the font size should be reduced.

 Table 3 needs correction (the data “has spread“ over 2 lines)

Throughout the entire text of MS there are no spaces after the numerical value when indicating degrees.

Conclusions. The text of the Сonclusion needs to be corrected, since it is very vague and largely duplicates the results. Please indicate the most important ideas of the work.

References. The list of references is not compiled according to the rules of MDPI journals, including Molecules.

Thus, there are some typos and inaccuracies in the MS presented. The review highlights only a few of them. Additionally, the quality of the MS would be significantly improved if the authors provided a graphical abstract.

Conclusion: I recommend the publication of the MS “Fractionation and Lability of Phosphorus Species in Cottonseed Meal-derived Biochars as Influenced by Pyrolysis Temperature” in Molecules Journal only after major revision.

Author Response

Many thanks for reviewing the manuscript Molecules-2722414 entitled "Fractionation and lability of phosphorus species in cottonseed meal-derived biochars as influenced by pyrolysis temperature." All the constructive comments have been carefully addressed and incorporated in the revised version of the manuscript. Details are presented as follows.

Comments: 

The MS titled “Fractionation and Lability of Phosphorus Species in Cottonseed Meal-derived Biochars as Influenced by Pyrolysis Temperature” was prepared based on original research. The information presented is of scientific interest and practical significance. In my opinion, it corresponds to the theme of the Molecules Journal. The MS title reflects the content of the article. Generally, the work is easy to understand. However, I cannot recommend MS in the form presented for publication in Molecules because it contains some inaccuracies.

First of all, the structure of the article does not meet the requirements of the Molecules journal. According to the “Instructions for authors” in Molecules, the following sequence of sections is recommended: Introduction, Results, Discussion, Materials and Methods.

Response: The manuscript has been adjusted to follow the recommended content organization in the order of Introduction, Results and Discussion, Materials and Methods, and Conclusions.

Abstract. The abstract provides a fairly clear overview of the research. Information is presented in a logical sequence.

Response: Thanks. The abstract remains without further modifications.

Keywords. It is not recommended to include the words used in the title of MS.

Response: The keyword list was updated with terms different from those in the title.

Introduction. The relevance of the problem to which the research is aimed is shown. The rationale for scientific and methodological approaches to solving this problem is presented. However, there is no clear statement of the purpose of the study.

Response: The objectives were stated at the end of the Introduction section. To highlight the objectives, the sentence was modified as “The main objectives of the present study were to investigate the chemical transformation of P in CSM during pyrolysis, to evaluate the lability of P in CSM-derived biochars generated at varied pyrolysis temperatures, and to gain information of optimal pyrolysis conditions for converting CSM to biochar products with desirable P supply dynamics in agricultural and environmental applications.”  

P1, L. 32, 33, 43: It is recommended to use more commonly used mass units than megagrams (Mg), especially since the MS repeatedly mentions magnesium with the same symbol.

Response: Agree. The unit megagram (Mg) was replaced with ton, with additional specification to indicate 1 ton = 1,000 kg.

 Materials and Methods. In general, the methodological section is written in sufficient detail, but subsection 2.5 needs to be improved:

P4, L. 172. It was shown that “all laboratory experiments of sample characterization and analysis were conducted in duplicates”. P. 4, L. 179: “Quantitative results are expressed as mean ± standard deviation of duplicate measurements”. What the authors mean by “in duplicates measurements” – two independent experiments, data of which were averaged? How many biological and analytical replicates were in each measurement?

Response: In analytical chemistry and the present manuscript, a duplicate means a repetition of the entire experimental procedure and “duplicate measurements” refers to the data points obtained from the experimental duplicates. To eliminate the confusions, the term “duplicate measurements” was replaced with “experimental duplicates” in the texts and table/figure captions of the revised manuscript.

Results and Discussion. The authors presented quite a lot of Figures and Tables, which illustrate the work well. Figure 1 is of particular interest. However, there are a number of comments on the design of the illustrations.

First of all, if the number of repetitions of measurements differed, it is advisable to display this information in Tables and Figures (n = ?).

Response: “experimental duplicates” was used in table/figure captions to take place of the originally “duplicate measurements.”  In Materials and Methods section, “n = 2” is added to indicate experimental duplicates.

Table 1. The heading of the 1st column (“Product”) does not seem to be correct.

Response: “Product” was originally used to define the first row of the table. To avoid the confusion, the word was deleted from Table 1.

Figure 3. Сaption: “superscript letters denote significance of difference between treatment levels” is wrong.

Response: Thanks. The sentence was removed from Figure 3 caption.

In Figures 2–4 captions, the font size should be reduced.

Response: Thanks. The font size of all the figure captions was adjusted to 12 pts.

Table 3 needs correction (the data “has spread“ over 2 lines)

Response: Thanks. The font size was reduced to ensure the same line presentation of individual data points in the table.

Throughout the entire text of MS there are no spaces after the numerical value when indicating degrees.

Response: It is a standard American English format to leave no space between a number and the following degree symbol for indicating temperature or angle.

Conclusions. The text of the Ð¡onclusion needs to be corrected, since it is very vague and largely duplicates the results. Please indicate the most important ideas of the work.

Response: Thanks. The Conclusions section has been significantly shortened to skip the summary contents yet focus on the findings of P transformation and lability changes during thermochemical conversion of cottonseed meal to biochar.  

References. The list of references is not compiled according to the rules of MDPI journals, including Molecules.

Response: Thanks. The references were checked one by one with format adjustments to the MDPI Molecules styles.  

Thus, there are some typos and inaccuracies in the MS presented. The review highlights only a few of them. Additionally, the quality of the MS would be significantly improved if the authors provided a graphical abstract.

Response: The manuscript was thoroughly checked for data accuracy. Per the suggestion, a graphical abstract is included in the revised manuscript to illustrate the key findings.  

Reviewer 2 Report

Comments and Suggestions for Authors

The manuscript reports “Fractionation and lability of phosphorous species in CSM-derived biochars as influenced by pyrolysis temperature" submitted by Mingxin Guo et al. The manuscript is extensive and obtained biochars are quantified by using different extraction techniques followed by colorimetric P titrations. However, the authors need to address the following comments before being considered for publication.

1) How does CSM-derived biochar improve soil health? Please explain briefly.

2) Also author failed to explain the importance of biochar for fractionating the inorganic P.

3) The pyrolysis temperature almost took 141-445 min at 20 degrees/min But how long the CSM material was kept at the desired temperature? How does the author determine that complete pyrolysis has occurred?

4) From Table 2, as the pyrolysis temperature increases from 300 to 450 degrees the OP is around 10-14 % of TP and then a drastic increase was observed to 44% of TP from 450 to 600 degrees. The IP was decreased from 89 % TP to 56 % of TP. Why there was a drastic difference in the OP content with an increase in temperature? 

5) Based on the above observation, 1M HCl extractability of CSM was behaved. It is well correlated with the amount of IP. Please explain the details in the manuscript.

6) The author failed to provide how IP and OP were determined.

7) Can use different concentrations of HCl as well the different acidic strengths like H2SO4, and HNO3?

8) The conclusions need to be more concise without missing specific details. 

Comments on the Quality of English Language

English Editing is necessary

Author Response

Many thanks for reviewing the manuscript Molecules-2722414 entitled "Fractionation and lability of phosphorus species in cottonseed meal-derived biochars as influenced by pyrolysis temperature." All the constructive comments have been carefully addressed and incorporated in the revised version of the manuscript. Details are presented as follows.

Comments: 

The manuscript reports “Fractionation and lability of phosphorous species in CSM-derived biochars as influenced by pyrolysis temperature" submitted by Mingxin Guo et al. The manuscript is extensive and obtained biochars are quantified by using different extraction techniques followed by colorimetric P titrations. However, the authors need to address the following comments before being considered for publication.

1) How does CSM-derived biochar improve soil health? Please explain briefly.

Response: Biochar in general has the potential for improving soil health through ameliorating soil physical, chemical and biological properties if appropriately applied. The mechanisms through which biochar as a soil amendment can enhance soil health are reviewed in the paper “Guo, M. 2020. The 3R principles for applying biochar to improve soil health. Soil Syst. 4, 9. doi: 10.3390/soilsystems4010009.” This paper was additionally referenced in the revised manuscript via the statement “Agricultural application of CSM-derived biochar in place of raw CSM would achieve persistent soil health improvement effects through ameliorating soil physical, chemical, and biological properties [15].”

2) Also author failed to explain the importance of biochar for fractionating the inorganic P.

Response: During pyrolysis, organic P in biomass feedstock such as CSM is mostly transformed into inorganic P, resulting in biochar products with predominant inorganic P species. The inorganic P, existing in various associations in biochar, is commonly fractionated via sequential extraction techniques into different pools demonstrating varying mobility, lability, and plant availability. The importance of biochar P (predominantly inorganic P) fractionation is indicated in the manuscript through significant texts and Figures 2 and 3 elucidating the relative proportion, lability, and plant availability of readily labile (water-extractable), generally labile (NaHCO3-extractable), moderately labile (NaOH-extractable), low labile (HCl-extractable), and residual (unextractable) P fractions.

3) The pyrolysis temperature almost took 141-445 min at 20 degrees/min But how long the CSM material was kept at the desired temperature? How does the author determine that complete pyrolysis has occurred?

Response: It took 15-30 minutes to raise the oven temperature at 20°C/min from the ambient level (e.g., 20°C) to the desired temperature (e.g., 300–600°C). The pyrolysis peak temperature was maintained until the pyrolysis reaction was complete, as indicated by no further visible smokes emitted out of the oven. In the manuscript the pyrolysis operation is briefly described as “The furnace temperature was increased from the ambient temperature to the selected peak temperature at 20°C/min and maintained at the peak value until the pyrolysis reaction (pyrolytic decomposition) in the container was complete, as indicated by no further visible smokes being emitted out of the furnace [16].”

4) From Table 2, as the pyrolysis temperature increases from 300 to 450 degrees the OP is around 10-14 % of TP and then a drastic increase was observed to 44% of TP from 450 to 600 degrees. The IP was decreased from 89 % TP to 56 % of TP. Why there was a drastic difference in the OP content with an increase in temperature? 

Response: The IP proportions in biochars were estimated by 1 M HCl extraction, while the OP proportions were calculated from the IP measurements. As recalcitrant P minerals that are not extractable by 1 M HCl formed at high (e.g., >450°C) temperature pyrolysis, the IP proportions of high temperature biochar products was probably underestimated while OP proportions overestimated using the analytical methods. The apparently “abnormal” trends of biochar IP and OP proportions shifting as a function of the pyrolysis temperature are justified in the manuscript as “At a pyrolysis temperature 500°C and above, recalcitrant, less extractable phosphate minerals (e.g., hydroxyapatite and oxyapatite) formed with evident detection in PL-derived biochars, leading to the apparently “decreasing” proportion of IP estimated by 1 M HCl extraction and the “increasing” proportion of OP calculated from the differences between TP and IP in the biochar products [18]. In the present study, the less extractability of ash minerals in CSM-derived biochars at higher pyrolysis temperature (i.e., C500, C500, and C600) is evidenced by the increasing mineral ash contents yet decreasing EC values (Table 1). It is postulated that most (e.g., >90%) of the OP in CSM was transformed into IP during pyrolysis at ≥500°C.”

5) Based on the above observation, 1M HCl extractability of CSM was behaved. It is well correlated with the amount of IP. Please explain the details in the manuscript.

Response: Again, commonly accepted is that IP in environmental samples is extractable by 1 M HCl and can be quantified by acid extraction. Unfortunately, recalcitrant phosphate minerals such as hydroxyapatite and oxyapatite are not extractable by 1 M HCl. The results of IP proportions in CSM-derived biochars were obtained using the HCl extraction techniques but might not discover the true situations.

6) The author failed to provide how IP and OP were determined.

Response: The analytical methods were briefly described in the manuscript as “Aliquots of the raw CSM and the biochar samples were further ground to <0.15 mm prior to P characterization. The TP content was determined following colorimetric P measurement of acid digested samples [18]. The TP consists of inorganic P (IP) and organic P (OP) in most environmental matrices. The IP content of the samples was quantified using 24-hr 1 M HCl extraction followed by colorimetric P measurement of the extracts [18]. The difference between the TP and IP contents of a sample was computed as the OP content.”

7) Can use different concentrations of HCl as well the different acidic strengths like H2SO4, and HNO3?
Response: 1 M HCl is the research proved and commonly used agent for effectively extracting inorganic P from environmental matrices. A lower concentration compromises its extractability; a higher concentration may not be necessary as a result of the increased costs and the reduced H+ activity. Relative to HCl, H2SO4 and HNO3 are more oxidative and corrosive. Furthermore, many sulfate minerals demonstrate low solubility in water. It is practical to enhance the extractability of 1 M HCl by reducing the sample particle size, increasing the extraction temperature, and elongating the extraction time.

8) The conclusions need to be more concise without missing specific details. 

Response: Thanks. The Conclusions section has been significantly shortened to skip the summary contents yet focus on the important findings of P transformation and lability changes during thermochemical conversion of cottonseed meal to biochar.  

Round 2

Reviewer 1 Report

Comments and Suggestions for Authors

Dear authors!

In my opinion, after revision, the quality of the article has noticeably improved. The authors worked on the MS, changed the structure in accordance with the journal template, improved the quality of Figures and took into account some other comments. However, some editorial edits are required.

Few comments and remarks are listed below but they are not exhaustive:

1. It is necessary to correct the indices of the symbols and units of measurement (must be superscript). For example:

P. 2, L. 85-86; P. 5, L. 212, 218, 221, 224, 228, 231; P. 8, L. 274-275; P. 9, L. 308, 313; P. 15, L. 466, 467; P. 16, L. 525 and so on.

2. Table 3 needs correction. Indexes in the left column and values in lines are shifted to another paragraph. Maybe it makes sense to reduce the font size?

3. It is necessary to give the explanation of the designations at the first mention in the text rather than in section 3, which follows later.

For example: OC (L. 205), EC (L. 212), TN (L. 221), TP (L. 227), SEM (L. 259), Pir, Pix, Po (L. 347) and so on.

4. Additionally: P. 5, L. 212 – It’s not entirely correct to decipher ES as Salinity level.

5. There is no uniformity in the spelling of some abbreviations. For example: Pir, Pix, Po in the text and below the Table 3.

Conclusion: I recommend the publication of the MS “Fractionation and Lability of Phosphorus Species in Cottonseed Meal-derived Biochars as Influenced by Pyrolysis Temperature” in Molecules Journal after minor revision.

Author Response

The manuscript has been revised by incorporating all the comments and suggestions. Please find attached the updated version of the manuscript for details in red. Thanks.

Reviewer 2 Report

Comments and Suggestions for Authors

It is very tough navigating a manuscript with track changes, especially when clarity is crucial. Properly communicating with the reviewers during the reviews is key. Please be considerate of the reviewer's time.

Accept in current form

Comments on the Quality of English Language

Moderate English editing is necessary.

Author Response

(The authors gave the same response as above.)
